

# Anticorrelation of Net Uptake of Atmospheric $CO_2$ by the World Ocean and Terrestrial Biosphere in Current Carbon Cycle Models

Stephen E. Schwartz

School of Marine and Atmospheric Sciences
Stony Brook University, Stony Brook NY 11794 USA

*Correspondence to*: Stephen.Schwartz@stonybrook.edu

**Abstract**. The rate at which atmospheric carbon dioxide ($CO_2$) would decrease in response to decrease of anthropogenic emissions or cessation (net zero emissions) is of great scientific and societal interest. Such decrease in atmospheric $CO_2$ on

the centennial scale would be due essentially entirely to transfer of carbon into the world ocean (WO) and the terrestrial biosphere (TB), which are sink compartments on this time scale. The rate of decrease of excess atmospheric $CO_2$ and the apportionment of this decrease into the two sink compartments has been examined in two prior model intercomparison studies, subsequent either to a pulse emission of $CO_2$ or to abrupt cessation of anthropogenic $CO_2$ emissions. The present study examines and quantifies inter-model anticorrelation in those studies in the net rate and extent of uptake of $CO_2$ into the

two sink compartments. Specifically, in each study the time-dependent coefficients characterizing the net transfer rate into the two sink compartments, (evaluated as the net transfer rate normalized to excess atmospheric $CO_2$ above pre-pulse amount, for the pulse experiment; or as the net transfer rate divided by excess atmospheric $CO_2$ above preindustrial amount, for the abrupt cessation experiment) was found to exhibit strong anticorrelation across the participating models. That is, models for which the normalized rate of uptake into the WO was high exhibited low uptake rate into the TB, and vice versa. This

anticorrelation in net transfer rate results in anticorrelation in net uptake extent into the two compartments that is substantially greater than would be expected simply from competition for excess $CO_2$ between the two sink compartments. This anticorrelation, which is manifested in diminished inter-model diversity, can lead to artificially enhanced confidence in current understanding of the consequences of potential future reductions of $CO_2$ emissions and in the global warming potentials of non-$CO_2$ greenhouse gases relative to that of $CO_2$.

**1 Introduction**

Essentially the only means by which excess carbon dioxide ($CO_2$), i.e., above preindustrial, is removed from the atmosphere on the centennial scale is net uptake by the world ocean (WO) and the terrestrial biosphere (TB). Here the term net uptake is used to emphasize that transfer of $CO_2$ from the atmosphere to these two compartments of Earth's biogeosphere is highly



reversible; hence it is this net transfer that is pertinent to the disposition of excess atmospheric $CO_2$ due to anthropogenic

emissions on this time scale. Historically, over the industrial era, the increase in atmospheric $CO_2$ has been about half of anthropogenic emissions (Bacastow and Keeling, 1973; Friedlingstein et al., 2022). Hence the processes governing this net uptake, and model-based representation of these processes, are of great scientific and societal interest. The need for confident understanding and model based representation of these processes has become increasingly acute because of the societal need to assess the consequences of prospective reduction or cessation (net zero emissions) of anthropogenic $CO_2$ emissions and to

develop effective strategies to achieve specified targets, such as mixing ratio of atmospheric $CO_2$ or increase in global temperature.

A key means of assessing current understanding and model-based representation of processes governing the drawdown of atmospheric $CO_2$ in carbon-cycle models is model intercomparison studies, more specifically comparison of the modeled response of the change of the amount of carbon in the atmosphere, the WO, and the TB in response to a perturbation in

emissions. (A perturbation in emissions is required because in the absence of such a perturbation the carbon stocks in the three compartments would be in steady state, i.e., not changing.) Such intercomparison studies are intended to reveal, through inter-model diversity in model response, the extent to which model-based representations of the processes governing net uptake of $CO_2$ differ among the participating models, and thus to serve as a measure of uncertainty in understanding. In principle as well such intercomparisons might reveal reasons for differences among the models and lead to improvement in

areas of disagreement.

In this study attention is directed to two key model intercomparison studies that examined changes in carbon stocks in the three compartments (atmosphere, WO, and TB) following abrupt changes in $CO_2$ emissions. The three compartments are considered a closed system; consequently the net change in atmospheric $CO_2$ subsequent to the pulse (or cessation) is the complement of the net changes in carbon stocks of the other two compartments. Joos et al. (2013; hereinafter J13) reported

the disposition of a pulse emission of $CO_2$ of magnitude 100 Pg C emitted into the atmosphere as amount of increase of carbon in each of the three compartments as a function of time subsequent to the emission pulse, thereby obtaining impulse response functions (IRFs; fraction of emitted pulse in each compartment as a function of time subsequent to the pulse) for each of the three compartments. MacDougall et al. (2020; hereinafter M20), examined the decrease in atmospheric $CO_2$, and the net uptake into the WO and the TB, subsequent to abrupt cessation of anthropogenic $CO_2$ emissions (ZECMIP study,

Zero Emissions Commitment Model Intercomparison Project). The two studies are not wholly independent in that several models or modeling groups participated in both intercomparisons.

Because these two studies reported not just the decrease in atmospheric $CO_2$ but also net uptake of carbon by the WO and the TB, it is possible to assess inter-model diversity not just in the decrease in atmospheric $CO_2$ but also in the net changes in the carbon stocks in other two compartments in response to the perturbation in emissions, as governed by the process represented

in the models. Here it should be noted that some inherent anticorrelation across the models is expected in the extent of uptake



into the WO and the TB; because the WO and the TB are the only sinks of atmospheric $CO_2$, a greater uptake into one compartment would necessarily result in a lesser uptake into the other. In contrast, the transfer coefficients characterizing the net uptake of atmospheric $CO_2$ into the WO and the TB, defined as the time-dependent net rate of uptake into the compartment divided by the time-dependent amount of excess atmospheric $CO_2$, would be expected to be uncorrelated if

treatment of the processes governing the net uptake rate into one compartment were independent from that governing the net uptake rate into the other compartment across the set of models participating in the studies. Here the inter-model correlations in extent and rate of uptake of atmospheric $CO_2$ are examined to assess the independence of the representation of uptake of atmospheric $CO_2$ into the WO and the TB in the models participating in each of the two intercomparison studies. Any substantial negative correlation transfer coefficients representing uptake of atmospheric $CO_2$ into the two compartments

would be indicative of compensating effects, whereby models having net uptake rate into one compartment have higher net uptake rate into the other, thereby resulting in a reduced inter-model diversity in the rate and extent of decrease of atmospheric $CO_2$. As inter-model diversity is commonly taken as a measure of uncertainty, any such negative correlation would artificially reduce the perceived uncertainty in the rate of drawdown of excess atmospheric $CO_2$ subsequent to a perturbation.

**2 Analysis and findings**

**2.1 Joos et al. (2013*)***

J13 examined and reported the extent of decrease of atmospheric $CO_2$ and the integrated net flux from the atmosphere into the ocean and into the TB subsequent to a 100 Pg-C emission pulse added to an atmosphere having prior $CO_2$ mixing ratio of 389 ppm as the difference between the run with the 100 Pg pulse and a reference run with mixing ratio held constant at 389

ppm. Participating models were three comprehensive Earth System Models (ESMs), seven Earth System Models of Intermediate Complexity (EMICs), and four box-type models. The disposition of excess atmospheric carbon was presented graphically as IRFs, the fractions of the emitted pulse in the atmosphere, the ocean and the TB, over two time periods, 0.5 to 100 years subsequent to the pulse and 100 to 1000 years; here results are examined only for the first 100 years, over which time the excess atmospheric stock resulting from the pulse decreased by about 60%. The J13 study has assumed considerable

importance because its multimodel mean is widely used for evaluation of the consequences of prospective future profiles of $CO_2$ emissions and for comparison of integrated radiative forcing of non-$CO_2$ greenhouse gases with that of $CO_2$ (so-called greenhouse warming potentials; e.g., Myhre et al., 2013, IPCC Fifth Assessment Report).

The time dependence of the IRFs for the several compartments over the 100-year time period following pulse emission, **Fig. 1*a***, shows that all models exhibited monotonic or nearly monotonic net increase in ocean IRF over this period, with the IRFs

of the several models relatively smooth over the time period. In contrast several of the models exhibited substantial fluctuations in extent of net uptake into the TB, **Fig. 1*b***; a negative derivative denotes a temporary negative net flux, i.e., net





flux from TB to atmosphere. The atmospheric IRFs, nominally the complements of the sum of ocean and TB IRFs, exhibited fluctuations more or less comparable to those of TB IRF for the several models; here a positive slope denotes a temporary increase in atmospheric stock. Comparison of the inter-model diversity of the IRFs (as shown by the ranges of pink shading denoting ± 2 s.d.) on the same vertical scale for the several IRFs demonstrates that the inter-model diversity of the atmospheric IRFs, **Fig. 1c**, to be systematically and substantially lower than that for the TB. As the decrease of the atmospheric stock is the sum of the uptakes into the ocean and TB compartments, the variance across the models of the atmosphere IRFs would, if uptake into the ocean and TB were uncorrelated across the model, be expected to exceed the variance for both the ocean and TB IRFs. The contravention of that expectation must therefore be due to anticorrelation of the IRFs across the models. This anticorrelation is examined further in **Fig, 1d**, by means of the regression slope of IRF(WO) vs IRF(TB), which is negative throughout essentially the entire 100-year time period, approaching -0.4 by the end of the time period, with the difference from the value 0, which would be expected in the absence of anticorrelation of the regression slope up to several fold greater than the s.d. of that slope obtained from the linear least-squares regression. (Throughout this study correlation coefficients, regression coefficients, and their uncertainties are calculated by the linear regression model, as described, e.g., by Press et al., 1997.) The anticorrelation is examined further in **Fig, 1e**, which shows the Pearson $r$ correlation coefficient to be negative throughout the time period, approaching -0.9 at the end of the time period, again indicative of strong anticorrelation of IRF(WO) and IRF(TB) across the set of models that participated in this intercomparison study.

An example of the anticorrelation across the suite of models between the IRFs for uptake into the ocean and the TB is shown in **Fig. 2a** for time subsequent to pulse injection, time $t$ = 99.5 years. This example likewise shows strong anticorrelation, with models exhibiting high uptake into the ocean exhibiting low uptake into the TB, and vice versa. The Pearson $r$ coefficient is negative, with value -0.85, and $r^2$, the fraction of the variance in the relation between IRF(WO) and IRF(TB) accounted for by the regression is 0.72. The standard deviations of the IRFs obtained with the several models at $t$ = 99.5 yr, are 0.061 and 0.118, respectively, much greater for the TB than for the ocean, as shown also in **Fig. 1 a and b**. Using these values for $\sigma_O$ and $\sigma_T$ yields an unbiased estimate of $\sigma_A$ at $t$ = 99.5 yr, evaluated as the quadrature sum,

$$\sigma_A = \left( \sigma_O^2 + \sigma_T^2 \right)^{1/2} = 0.132$$ , nearly twice the value obtained from the modeled atmospheric IRFs themselves, 0.071.

What are the causes of the anticorrelation? It would seem that there are two potential contributions to this. First would be competition between the two sinks for the excess atmospheric $CO_2$ remaining from the pulse at a given time subsequent to the pulse; if, at a given time, more of this excess $CO_2$ has previously been drawn down into one compartment, there would be less $CO_2$ available to be drawn down into other compartment, resulting in anticorrelation in the amounts in the two compartments, simply because of this competition effect. A second, more intrinsic and concerning source of anticorrelation would be compensating treatments in the several models of the processes that govern the uptake of the excess $CO_2$ from the pulse emission by the WO and the TB, such that if the rate of uptake into the WO in a given model were high relative to the other models, the rate of uptake into the TB would be low, and vice versa.



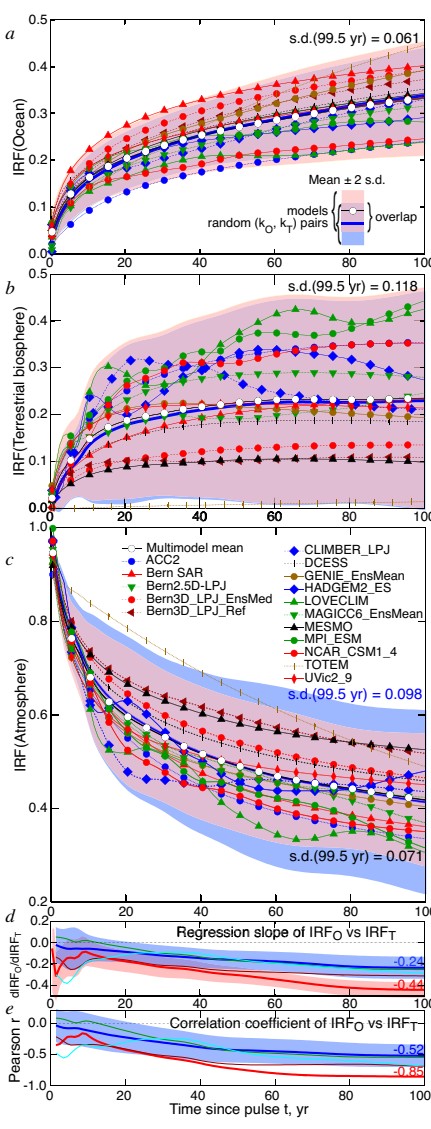


**Figure 1**. Time dependent impulse response functions (IRF's, $f$) for the perturbations in stocks in $a$, the world ocean $f_O$; $b$, the terrestrial biosphere $f_T$; and $c$, the global atmosphere $f_A$, as a function of time $t$ subsequent to emission





of a 100-Pg(C) pulse of $CO_2$ into the atmosphere, in excess of the amount of carbon that would be present in each compartment for atmospheric $CO_2$ held constant at the value at the time of emission of the pulse, as calculated by

the models that participated in the model intercomparison study of Joos et al. (2013); for details of the intercomparison and identification of the models see J13; the three panels are to the same scales. Adapted from Joos et al. (2013). Pink shading denotes ± 2 s.d. about the multimodel mean; s.d.'s at $t$ = 99.5 years after pulse are in black text; thick blue curve and blue shading denote the mean and ± 2 times the mean s.d. of 120 instances of IRFs calculated with randomly sorted pairs of net transfer coefficients ($k_O$, $k_T$) taken from the sets of net transfer

coefficients evaluated from the data presented by J13; mean s.d. for $f_A$ at $t$ = 99.5 yr in blue text in $c$. Key at lower right of $a$ shows mean (black, white markers) ± 2 s.d. for models (pink shading), mean (thick dark blue) ± 2 s.d. (light blue shading) for randomly sorted ($k_O$, $k_{T,i}$) pairs, and region of overlap (violet shading). Attention is called to slight non-overlapped regions in $a$ and $b$ and large non-overlapped region in $c$. $d$, Thick red, regression slope of linear fit of $f_O$ vs $f_T$ among the models for each year over the 100-year period subsequent to pulse emission

(example shown in **Fig. 2**); pink shading denotes estimated error in slope as obtained from the regression (±1 s.d.); thick blue, mean of regression slope of linear fit of $f_O$ vs $f_T$ evaluated for 120 instances of randomly sorted ($k_O$, $k_{T,i}$) pairs; blue shading denotes ± 1 s.d. of these 120 instances; thin curves denote time series of regression slopes for three instances of such random pairings. $e$, Pearson correlation coefficient $r$ of regression of $f_O$ vs $f_T$ for the several models, thick red; thick blue and blue shading, mean and s.d. of $r$ for 120 instances of IRFs, as in $d$; thin

curves denote correlation coefficients for three instances of those regressions.

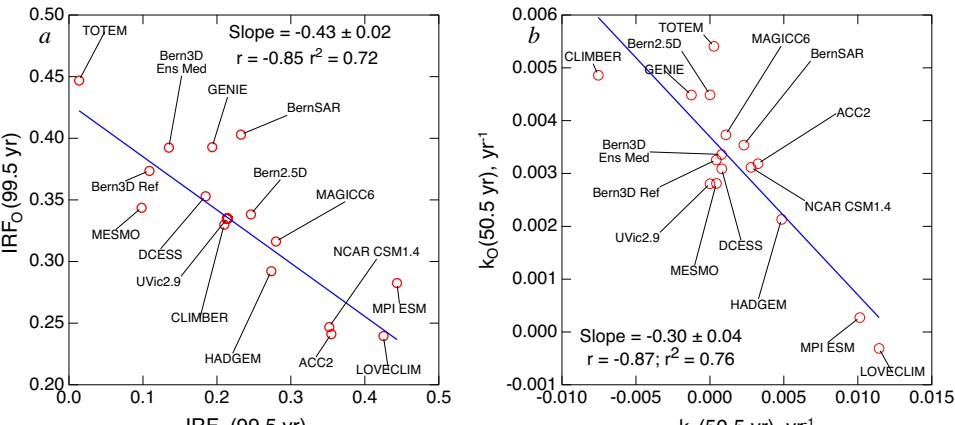

**Figure 2**. $a$. IRF for uptake of excess $CO_2$ by the world ocean versus that by the terrestrial biosphere, both given as fraction relative to the magnitude of the pulse, at time $t$ = 99.5 years subsequent to pulse injection, as



represented by the models that that participated in the model intercomparison study of Joos et al. (2013, J13). Blue line denotes linear regression; also shown are the regression slope; the Pearson correlation coefficient $r$, the negative value of which denotes anticorrelation; and $r^2$, the fraction of the variance that is accounted for by the regression. For identification of models see J13. *b*. As in *a*, but for transfer coefficient for net uptake of excess $CO_2$ from the atmosphere to the WO $k_O$ versus that from the atmosphere to the terrestrial biosphere $k_T$ at time $t =$

50.5 yr; negative values denote net transfer from the ocean or the TB to the atmosphere at that time.

Assessment of whether, and to what extent, the anticorrelation is due to the treatment of the processes versus due simply to competition requires examination of the rates of transfer of excess $CO_2$ as a function of time subsequent to the pulse, rather than of the IRFs themselves, which are the integrals of the transfer rates. The IRFs would be subject to the competition effect, whereas the time-dependent transfer rates would not. Hence, to assess the extent to which the anticorrelation is due to the

competition effect a similar analysis was conducted of the coefficients characterizing the rate of net transfer of excess $CO_2$ from the atmosphere into each of the sink compartments. To this end the net transfer coefficients into the ocean and the terrestrial biosphere were evaluated as the rate of net uptake of $CO_2$ (reckoned as C) into the specific compartment (WO or TB) per excess stock remaining in the atmospheric compartment at a given time:

$$k_O = \frac{df_O / dt}{f_A} \text{ and } k_T = \frac{df_T / dt}{f_A} \qquad (1)$$


respectively, where $f_O$ and $f_T$ denote the time-dependent impulse response functions (IRFs) of the two compartments and $f_A$ similarly denotes the time dependent IRF of the atmospheric stock, all expressed as fractions $f$ of the carbon mass of the emitted pulse. Dividing the time-dependent net transfer rates ($df_O/dt$, $df_T/dt$) by the time-dependent atmospheric IRF ($f_A$) to obtain the time-dependent net transfer coefficients ($k_O$, $k_T$) results in quantities that would be constant if the net fractional

uptake rate of excess $CO_2$ exhibited a constant proportionality to the amount of excess $CO_2$, much like a rate constant for a first-order reaction in chemical kinetics. The departures from constancy in $k_O$, **Fig. 3a**, and to much greater extent in $k_T$, **Fig. 3b**, are manifestations of the effects of the history of prior uptake of $CO_2$ by the two compartments and of the time-dependent states of these compartments affecting this net uptake.

Similarly the coefficient denoting the net rate of decrease of the IRF for stock in the global atmosphere is given by

$$k_A = -\frac{df_A / dt}{f_A} ; \qquad (2)$$


the derivatives were calculated numerically from the IRFs presented by J13. (As the figures of J13 were published as vector graphics, it was possible to digitize the data quite precisely; the data are also tabulated at



https://climatehomes.unibe.ch/~joos/IRF_Intercomparison/). The results of this examination are presented in **Fig, 3** as a function of time subsequent to the pulse and in **Fig. 2b** for a single value of time. At first look, the transfer coefficients plotted similarly to the IRFs are similar to the IRFs themselves. However the values of the $k$'s, being proportional to the derivatives of the IRFs, exhibit much greater fluctuations than the IRFs themselves (especially $k_T$), some being rather smooth, others exhibiting rather large, high frequency fluctuations. A local maximum in the IRF results in a change in sign (from positive to negative) in the net transfer coefficient; that is net transfer of excess $CO_2$ resulting from the emission pulse being transferred thereafter from the WO or the TB back to the atmosphere; and correspondingly a local minimum in the IRF resulting in a change in sign in the transfer coefficient from negative to positive. Essentially no such return transfer from the ocean to the atmosphere is exhibited by the any of the models (very slight return transfer for two of the models); in contrast, all but three of the models exhibited return flux from the TB to the atmosphere at one or more times in the 100-yr period examined, **Fig. 3b**, with two of the models exhibiting instances of return flux of greater than 1% $yr^{-1}$ relative to the amount of remaining excess atmospheric stock. As with the IRFs themselves, the anticorrelation between the transfer coefficients $k_O$ and $k_T$ is demonstrated graphically by the narrower spread of $k_A$ than $k_T$; **Figs. 3a-c** are drawn to the same scale. Quantitatively, as manifested by the traces for regression slope and correlation coefficient, throughout the time over which the two compartments were drawing down the major fraction of the pulse-emitted $CO_2$, roughly the first 60 years or so subsequent to the pulse, **Fig. 1c**, there is again systematic anticorrelation, here between $k_O$ and $k_T$ (negative of Pearson $r$ as great as 0.88). An example, for time $t = 50.5$ yr subsequent to the pulse, **Fig. 2b**, shows quite strong anticorrelation in the transfer coefficients into the WO and the TB across the models, $r^2 = 0.72$. This anticorrelation in the transfer coefficients themselves rules out the anticorrelation in the IRFs as being due solely to the competition effect and demonstrates that for models for which the transfer rate into the ocean was high relative to the multimodel mean, the transfer rate into the TB was low relative to the multimodel mean, and vice versa.

The cause of the anticorrelation of the IRFs was further assessed by calculations of the IRFs in which the $k_T$'s of the several models were randomly paired with the $k_O$'s, with examination of the resultant correlations between IRFs for uptake by the WO and the TB. First, the possibility of any correlation among the random pairs of the time dependent transfer coefficients was assessed; as anticipated, these random pairings exhibited essentially no correlation. The mean regression slope over the 120 instances was essentially zero over the 100 yr period; mean at $t = 100$ yr -0.0009 (**Fig. 3d**); and the mean Pearson $r$ was likewise essentially zero, with standard deviation about 0.25 over the time range, with mean at $t = 100$ yr -0.003 (**Fig. 3e**). To be sure there was meandering of both quantities for individual instances of pairing, three of which are shown in the figures, with some of these individual curves meandering considerably outside ± 1.s.d., but that is of course to be expected, and none exhibited the systematic strong anticorrelation exhibited by IRFs of the models, as shown by the thick red curve, with -$r$ as great as 0.88, or 3.5 times the s.d. obtained with the randomly paired $k_O$ and $k_T$.





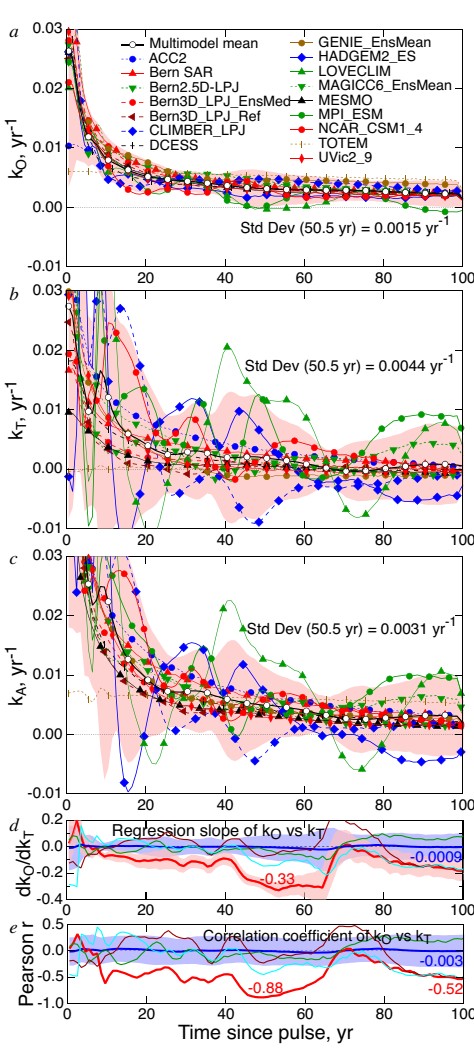

**Figure 3**. Net transfer coefficients from the atmosphere into $a$, the world ocean $k_O$ and $b$, the terrestrial biosphere $k_T$ as a function of time $t$ subsequent to emission of a 100-Pg (C) pulse of $CO_2$ into the atmosphere, evaluated from the results from the model intercomparison of Joos et al., 2013; similarly, $c$, the fractional rate of net decrease of





excess $CO_2$ stock in the atmosphere; negative value denotes temporary net increase. Pink shading denotes ± 2 s.d. about the multimodel mean; s.d.'s at $t = 50.5$ yr are indicated; the three panels are to the same scale. For

identification of the models see J13. *d*, Regression slope, thick red, of linear fit of $k_O$ vs $k_T$ for the several models for each year over the 100-year period subsequent to pulse emission (see **Fig. 2*b*** for example), with value at $t = 53.5$ yr indicated, red text; pink shading denotes ± 1 s.d.; mean regression slope, thick blue, and ± 1 .s.d., blue shading, for 120 instances of $k_O$ and $k_T$ of different models being randomly paired, with mean value at $t = 100$ yr indicated, blue text; thin curves denote time series of three instances of those regression coefficients. *e*, Pearson

correlation coefficient *r* of linear regression of $k_O$ vs $k_T$ for the several models, thick red, with values at $t = 49.5$ and 99.5 yr indicated, red text; blue and blue shading, mean and ± 1 s.d. of *r*, for 120 instances of $k_O$ and $k_T$ of different models being randomly paired, with mean value at $t = 100$ yr indicated, blue text; thin curves denote time series of three instances of those correlation coefficients.

To assess the contribution of the competition effect to the anticorrelation in the IRFs themselves, the randomized pairs of

time-dependent $k_O$ and $k_T$ were used to calculate time-dependent IRFs corresponding to those randomized ($k_O$, $k_T$) pairs. To do this, first the time-dependent IRF of the atmospheric stock was evaluated for each ($k_O$, $k_T$) pair as the integral

$$f_A(t) = \exp\left(-\int \left(k_O(t) + k_T(t)\right) dt\right). \tag{3}$$

Then the IRFs of the WO and the TB were evaluated for each ($k_O$, $k_T$) pair as the integrals of the transfer rates.

$$f_O(t) = \int k_O(t) f_A(t) dt \text{ and } f_T(t) = \int k_T(t) f_A(t) dt \tag{4}$$

The accuracy of this approach was demonstrated by examination of the IRFs calculated in this way for the original ($k_O$, $k_T$) pairs, which closely reproduced the original IRFs. The results of this examination are shown in **Fig. 1*d*** for the regression slope and **Fig. 1*e*** for the correlation coefficient of $f_O$ vs $f_T$ over the 100 yr period. In each panel the thick red curve denotes the time-dependent quantity evaluated for the ($k_O$, $k_T$) pairs of the 16 models of the Joos et al., study, as described above. The same analysis was then conducted for 120 instances in which a randomly sorted set of transfer coefficients $k_{T,i}$ was

paired with the set of transfer coefficients in the Joos et al., study $k_O$, and the model was run out for 100 years for each randomized pair of transfer coefficients ($k_O$, $k_{T,i}$); that process was then carried out for a total of 120 runs with randomized pairs ($k_O$, $k_{T,i}$), $0 \leq i \leq 119$. The thick blue curve denotes the mean of the respective quantity for the 120 randomized pairs ($k_O$, $k_{T,i}$) and the blue shading denotes the s.d. of the respective quantity. (As with the transfer coefficients, the curves for individual ($k_O$, $k_{T,i}$)) pairs meandered about the mean.) The non-zero values of the mean regression slope and the mean

correlation coefficient of the IRFs evaluated with the randomized pairs of transfer coefficients are measures of the competition effect, the negative regression coefficient *-r* increasing over the 100 year period, to 0.52 by the end of that period, a substantial fraction of the value for the models themselves, 0.85. The enhancement of the anticorrelation in IRFs



due to anticorrelation of the transfer coefficients may be inferred from the increase in the regression slope of $f_O$ vs $f_T$ for the original models compared to that for the randomized $(k_O, k_T)$ pairs, a factor of about 1.8.

The effect of anticorrelation between the net transfer coefficients in the models is examined also in **Fig. 1, *a-c***, by comparison of the standard deviations of the IRFs of the several compartments for each year up to 100 years following the pulse as calculated in model runs with 120 instances of the randomized pairs $(k_O, k_{T,i})$ with those of the models themselves. (Here the mean of the IRFs obtained from the integration, which was initiated at time subsequent to the pulse $t = 0$, was set to match the mean of the IRFs presented by J13 at $t = 0.5$ yr, the earliest time presented by J13.) The s.d.'s of the IRFs for the

WO and the TB for the randomized pairs, shown by the blue shading denoting $\pm$ 2 s.d. about the mean are essentially identical over the entire 100-year period to the s.d.'s for the models, shown by the pink shading, with nearly complete overlap (purple) between the blue and pink shading and only slight regions of non-overlap – pink toward the tops of the shaded areas in **Figs. 1*a* and *b***, and a slight region of non-overlapped blue toward the bottom of the shaded area in **Fig. 3*b***. However the s.d. for the IRF of the atmosphere calculated with the randomized pairs $(k_O, k_{T,i})$ is substantially greater than that for the

models, **Fig. 1*c***, as manifested by the substantial region of non-overlapped blue extending well above and below the overlap region. The decrease in inter-model spread of the atmospheric IRF of the models participating in the intercomparison relative to that calculated for random pairing of the transfer coefficients is thus confidently attributed to anticorrelation of the sets of transfer coefficients across the models.

    By the several measures presented here the inter-model spread of the IRFs denoting the decrease of atmospheric $CO_2$

subsequent to injection of a pulse of $CO_2$ presented in by J13 would seem to be well less than would be expected in the absence of inter-model anticorrelation of the normalized net transfer rates of atmospheric $CO_2$ to the WO and the TB. This reduction would seem to be due to compensating treatments of the underlying processes that control this transfer, such that for models for which $k_O$ was high relative to the multimodel mean, the average value of $k_T$ was low relative to the multimodel mean, and vice versa.

**2.2 MacDougall et al. (2020)**

    The ZECMIP model intercomparison study (MacDougall et al., 2020; M20) was similar to that of Joos et al. (2013), except that it examined the drawdown of atmospheric $CO_2$ (and uptake of this atmospheric $CO_2$ by the world ocean and terrestrial biosphere) subsequent to an abrupt cessation of emissions, rather than in response to a pulse emission. The protocol of the M20 study was that atmospheric $CO_2$ was forced, in each of the models, to increase from its preindustrial value at 1% per

year until doubled $CO_2$ was reached, at which point anthropogenic emission was abruptly ceased. Participating models were nine ESMs, and nine EMICs. The paper itself and supplementary material reported stocks of excess carbon in the atmosphere, the WO, and the TB and net transfer rates from the atmosphere to the WO and the TB subsequent to the cessation. Results from that study are examined here similarly to the examination of the results from the study of J13. Data



were obtained from the supplementary material of M20 and at http://terra.seos.uvic.ca/ZEC/ (kindly provided as a single zip

file by Andrew MacDougall).

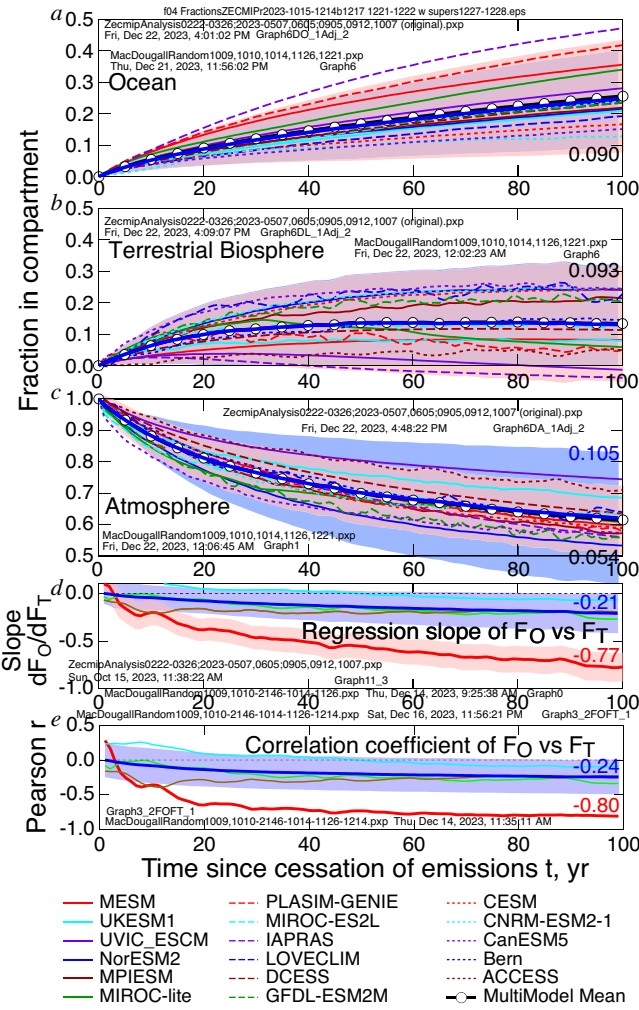

**Figure 4**. Time dependence of the fraction of excess atmospheric $CO_2$ (above preindustrial) at the time of cessation of anthropogenic emissions taken up by *a*, the world ocean, and *b*, the terrestrial biosphere, and *c*,





remaining in the atmosphere, subsequent to abrupt cessation of emissions, $f_O$, $f_T$, and $f_A$, respectively, as calculated by the models that participated in the ZECMIP intercomparison (MacDougall et al. 2020, M20). For details of the intercomparison and identification of the participating models see M20. Pink shading denotes $\pm$ 2 s.d. about the multimodel mean; s.d.'s at $t = 100$ yr after cessation in black text. Thick blue curve and blue shading denote mean and $\pm$ 2 times the mean s.d. of 120 instances of IRFs calculated with randomly sorted pairs of net transfer coefficients ($k_O$, $k_T$) taken from the set of net transfer coefficients evaluated from the data

presented by M20; violet shading denotes overlap of blue and pink shading as in **Fig. 1**. Mean s.d. for $f_A$ at $t = 100$ years is in blue text. The three panels are to the same scale. $d$, Regression slope of linear fit of time-dependent fraction of uptake into WO, $f_O$, vs into TB, $f_T$; pink shading denotes $\pm$ 1 s.d.; value at 100 yr in red text; thick blue, mean of regression slope of linear fit of $f_O$ vs $f_T$ evaluated for 120 instances of randomly sorted pairs of net transfer coefficients ($k_O$, $k_{T,i}$) taken from the set of net transfer coefficients evaluated from the data

presented by M20; blue shading denotes $\pm$ 1 s.d.; value at $t = 100$ yr in blue text; thin curves denote time series of regression slopes for three instances of such random pairings. $e$. Pearson correlation coefficient $r$ of linear regression of $f_O$ vs $f_T$ for the several models, thick red; mean and $\pm$ 1 s.d. of $r$, for 120 instances of $k_O$ and $k_T$, thick blue, as in $d$; values at $t = 100$ yr as in $d$; thin curves denote correlation coefficients for three instances of those regressions.

The decrease of excess $CO_2$ in the atmosphere and increases in the carbon stocks of the WO and the TB, subsequent to cessation of anthropogenic emissions, are all shown in **Fig. 4 *a-c*** as a fraction $f$ relative to the amount of excess $CO_2$ in the atmosphere at the time of cessation; all figures are at the same scale. The normalization to initial excess atmospheric $CO_2$, which accounts for differences among the models in the amount of excess atmospheric $CO_2$ at the time of cessation, permits comparison across the set of models of the apportionment of net transfer from the atmosphere, i.e., the decrease in $f_A$, from

its initial value of unity, into the WO, $f_O$, and the TB, $f_T$. The differences in the stocks in the WO and the TB relative to their values at the time of cessation, initially zero, increase as a function of time subsequent to cessation, as excess atmospheric $CO_2$ present at the time of cessation is taken up by each of those compartments in the absence of further anthropogenic emissions (i.e., net-zero anthropogenic emissions). This intercomparison is thus highly pertinent to a prospective transition to net-zero emissions, in contrast to the J13 study, which is pertinent to the disposition of an amount of $CO_2$ emitted at a given

time. Because of these different protocols the results of the two studies cannot not be directly compared. Importantly the initial changes in the stocks following the pulse emission, relative to the magnitude of the pulse (J13) are much greater than those following abrupt cessation of emissions, relative to the amount of excess atmospheric $CO_2$ at the time of cessation (M20).

For all the models excess atmospheric $CO_2$ began to decrease immediately upon cessation as excess atmospheric $CO_2$ was
being drawn into the WO and the TB. All models showed more or less monotonic decrease of atmospheric $CO_2$ throughout the initial 100 years subsequent to cessation; a few instances of temporary positive slope are indicative of slight, short-term



net increase in atmospheric $CO_2$. The fractional removal of excess atmospheric $CO_2$ at time subsequent to cessation $t = 100$ yr ranged across the models from 26% to 47% (i.e., fraction of excess $CO_2$ remaining 53% to 74%). All but one of the models showed the net amount of $CO_2$ taken up by the WO increasing monotonically throughout the initial 100 years. All models

also showed net uptake by the TB in the initial years subsequent to cessation, but with larger inter-model variation than for net uptake by the WO. In contrast to ocean uptake, the increase in TB amount exhibited substantial fluctuations for almost all the models. For all the models, toward the end of the 100 year period, the amount of uptake by the TB either leveled off, or decreased, or, in some models actually became negative (cumulative transfer from the TB back into the atmosphere over the initial 100 years exceeding cumulative transfer from the atmosphere into the TB). The spread of the extent of transfer into the

two individual compartments (0.090, 0.093 at $t = 100$ yr, 1 s.d.) was much greater than the spread in $f_A$ (0.054). The fraction of the excess atmospheric $CO_2$ at the time of cessation taken up by the ocean at 100 years $f_O(100$ yr) ranged across the models from 13% to 47%. The fractional uptake by the TB $f_T$ (100 yr) ranged from as great as about 25% in three models to negative values in two models (i.e., stock in the TB actually decreasing over the time period, by 1% or 4% relative to the initial excess atmospheric $CO_2$ ).

As in the results from the J13 intercomparison, there is indication of substantial anticorrelation across the suite of ZECMIP models between the extent of uptake into the WO and the TB. This is manifested in the first instance in the much narrower inter-model spread of the extent of decrease in the fraction of initial excess $CO_2$ remaining in the atmosphere than in the fractional uptake, again relative to the amount of excess atmospheric $CO_2$ at the time of cessation of anthropogenic emissions, by the ocean $f_O$ and the terrestrial biosphere $f_T$. This anticorrelation is manifested also in the Pearson correlation

coefficient, $r$, which is negative throughout almost the entire 100-year time period examined. The regression slope, initially rather small, increases (in magnitude) throughout the time period, almost to -0.8; a slope of -1 would indicate a one-to-one compensation between net uptake into the two compartments. This anticorrelation is manifested also in graphs of the net fractional uptake into the ocean versus that into the terrestrial biosphere, shown in **Fig. 5a**, at the end of the time period examined by M20, $t = 100$ yr after cessation of anthropogenic emissions.

Again, as with the results from J13, the question arises of the contribution to the anticorrelation across the models in extent of uptake of excess atmospheric $CO_2$ by the WO and the TB to competition between excess $CO_2$ taken up previously by one compartment reducing the amount available to be taken up by the other compartment. The contribution of this competition effect to the anticorrelation was first assessed by examining the net transfer coefficients, evaluated as the time-dependent rate of net increase in the fractional uptake into each of the two compartments divided by the time-dependent fraction of excess

atmospheric $CO_2$ , $k_O$ and $k_T$, **Fig. 6**. As the transfer rates and stocks were available from the supplementary materials and project website, it was not necessary to numerically differentiate the time dependent stocks, as was done in the analysis of J13. In several models the transfer coefficients were highly variable over time, with especially large fluctuations in the net transfer coefficients into the TB, panel $b$, and in the fractional rate of decrease of excess atmospheric $CO_2$, panel $c$, which is nominally the sum of the net transfer coefficients into the WO and the TB. Despite these large fluctuations, it was found that



the regression slope and $r$ value, evaluated for each year over the 100-year period following cessation, were both consistently negative, for almost every year in this time period. Importantly also, the correlation slope between $k_O$ and $k_T$ was negative throughout almost the entire time period. An example plot of $k_O$ vs $k_T$ is shown in **Fig. 5b**, for time $t = 55$ years subsequent to cessation, a time period during which fairly consistent negative correlation was exhibited, **Figs. 6d** and **6e**. While a single plot with correlation coefficient -0.58 might not by itself be convincing, the consistent negative correlation between $k_O$ and

$k_T$ would seem to make a fairly convincing case for negative correlation between these net transfer coefficients. Again, as was concluded for the models participating in the J13 intercomparison, such an anticorrelation would imply that models for which the terrestrial module resulted in relatively high uptake rate into the TB had ocean module that resulted in relatively low uptake rate into the WO, and vice versa. This anticorrelation between the transfer coefficients $k_O$ and $k_T$ resulted in substantial enhancement, by a factor of 3 or so, of anticorrelation of the net uptake extent into the two compartments beyond

that which would result only from the compensation effect as evaluated for random parings of $k_O$ and $k_T$, **Figs. 4d** and **4e**.

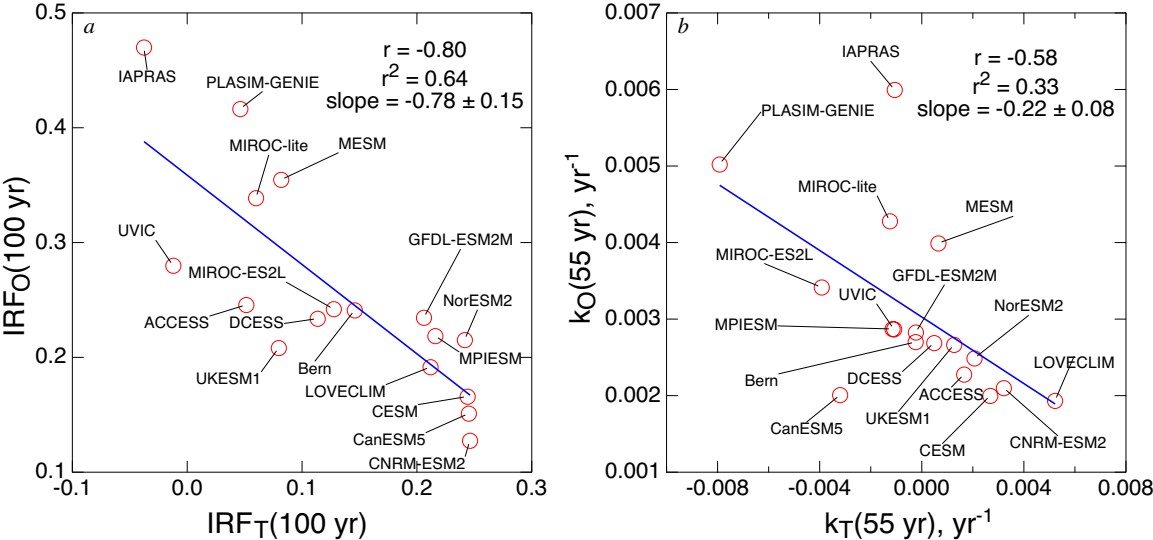

**Figure 5**. *a*. Net uptake of $CO_2$ in the world ocean $IRF_O$ versus that in the terrestrial biosphere $IRF_T$, both expressed as fraction of excess atmospheric $CO_2$ (i.e., above preindustrial) at the time of cessation of emissions, at time $t = 100$

years subsequent to cessation of anthropogenic emissions, as represented by the models that that participated in the ZECMIP intercomparison study of MacDougall et al. (2020, M20); negative values for $IRF_T$ denote net cumulative decrease in carbon stock in the TB over the 100-year time period. Blue line denotes linear regression; also shown are the regression slope, the Pearson correlation coefficient $r$, and $r^2$. For identification of models see M20. *b*. As in *a*, but





for transfer coefficient for net uptake of excess atmospheric $CO_2$ from the atmosphere to the WO $k_O$ versus that from the atmosphere to the TB $k_T$ at time $t$ = 55 yr; negative values of $k_T$ denote net transfer from the TB to the atmosphere at that time .

As with the analysis of J13, the cause of the anticorrelation extent of uptake into the WO and the TB was further assessed by calculations in which the transfer coefficients $k_T$ of the several models were randomly paired with the $k_O$'s with examination of the resultant correlations between uptake by the WO and the TB and the corresponding transfer

coefficients, 120 such random pairings. Again these random pairings exhibited essentially no correlation in the transfer coefficients themselves. The mean correlation slope $b$ at 55 years after cessation was -0.01 ± 0.11 (1 s.d.), (**Fig. 6d**), well less than the regression slope for the ($k_O$, $k_T$) pairs for the models, -0.22 ± 0.14. A similar situation was found for the correlation coefficient, **Fig. 6e**, $r$(55 yr) = 0.03 ± 0.30 vs -0.58 for the ($k_O$, $k_T$) pairs for the models. Again there was meandering of the correlation coefficient of individual instances of pairing, three of which are shown in the figure.

Some of the individual curves meandered considerably outside ± 1.s.d., which is of course to be expected, but none exhibited the systematic strong anticorrelation exhibited by IRFs of the models, as shown by the thick red curve.

The anticorrelation of the transfer coefficients was found to exert quite a substantial influence on the anticorrelation of the extent of transfer, **Fig. 4**. Here, in panels *a-c*, the curves for the original models and the associated spread, the pink shaded region representing ± 2 s.d. of the fraction of the excess atmospheric $CO_2$ at the time of cessation of anthropogenic emissions

that is transferred into the WO or the TB, and the fraction remaining in the atmosphere are superimposed on the averages taken over 120 random pairings of $k_O$ and $k_T$, thick blue line (and averages of 2 s.d., blue shading). For transfer extents $f_O$ and $f_T$ the range of the blue shading (randomized pairs ($k_O$, $k_{T,i}$)) is essentially the same as that of the pink shading (models). However for the fractional extent of removal from the atmosphere $f_A$, while the curve representing the mean of the 120 instances of random $k_O$, $k_T$ pairs (dark blue; no vertical adjustment, because time series started at $t$ = 0) is

virtually identical to the mean of the models themselves (black with open circles), the spread of $f_A$ for the randomly paired $k_O$ and $k_T$ (blue shading) substantially exceeds that for the models themselves, by nearly a factor of 2 by the end of the model runs. The difference between the blue shaded area (spread of the randomized pairs ($k_O$, $k_{T,i}$)) and the pink shaded area (spread of the models themselves) is a measure of the decrease in $f_A$ that is due to anticorrelation of the transfer coefficients in the models. The effect of anticorrelation in the transfer coefficients is seen also in the regression

slope $b$ of $f_O$ vs. $f_T$, -0.77 ± 0.15 for the models themselves vs. -0.21 ± 0.21 for the randomized pairs ($k_O$, $k_{T,i}$) and in the correlation coefficient $r$, -0.80 for the models vs. -0.24 for the randomized pairs (all at 100 years after cessation of anthropogenic emissions). Again, values of $b$ and $r$ for individual random ($k_O$, $k_T$) pairs meander more or less within the space of ± 1 s.d., but this meandering is well less that the separation between the mean quantities for the randomized pairs and the values for the models.






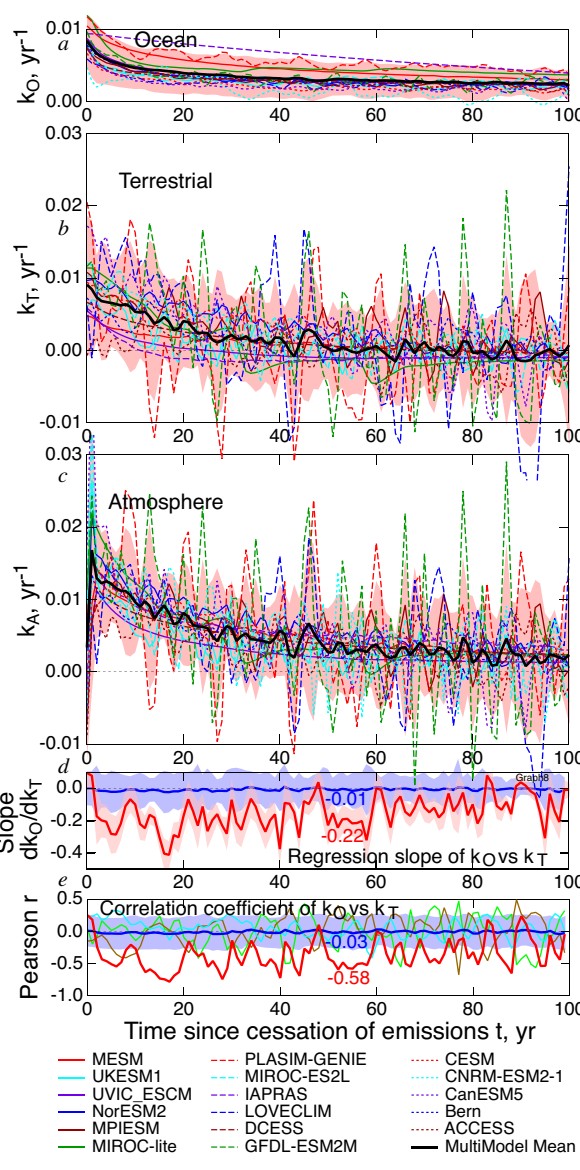

**Figure 6**. Net transfer coefficients from the atmosphere into *a* the world ocean $k_O$ and *b* the terrestrial biosphere $k_T$ as a function of time subsequent to cessation of emission of anthropogenic $CO_2$ into the atmosphere, evaluated from the results presented by MacDougall et al. (2020; M20); similarly, *c*, the fractional rate of



decrease of excess $CO_2$ stock in the atmosphere. Pink shading denotes $\pm$ 2 s.d. about the multimodel mean; the three panels are to the same scale. For identification of the participating models see M20. *d*, Regression slope, thick red, of linear fit of $k_O$ vs $k_T$ for the several models for each year over the 100-year period subsequent to cessation of emissions (see **Fig. 5*b*** for example), with value at 55 yr indicated; shading denotes $\pm$ 1 s.d.; mean regression slope, thick blue, and $\pm$ 1 .s.d., blue shading, for 120 instances of $k_O$ and $k_T$ of different models being

randomly paired, with mean value at 55 yr indicated; thin curves denote time series of three instances of those regression coefficients. *e*, Pearson correlation coefficient *r* of linear regression of $k_O$ vs $k_T$ for the several models, thick red, with value at 55 yr indicated; blue and blue shading, mean and $\pm$ 1 s.d. of *r*, for the 120 instances of random pairing, with mean value at 55 yr indicated; thin curves denote time series of three instances of those correlation coefficients.

**3 Discussion**

The intent of this study was to determine, for the two model intercomparison studies under examination, J13 and M20, the extent and causes of anticorrelation across the models participating in each study in the net uptake of excess $CO_2$ by the WO and the TB under the experimental protocols of the two studies. This study thus assessed not just anticorrelation across each of the model sets in the net uptake extent but also in the net transfer coefficients, the latter defined as the time-dependent rate of transfer into

the WO and TB divided by the time-dependent amount of atmospheric $CO_2$, for J13, in excess of the amount atmospheric $CO_2$, subsequent to the pulse, that would have been present in the absence of the pulse, and for M20, in excess of preindustrial $CO_2$ subsequent to cessation of anthropogenic emissions. As these net transfer coefficients are not dependent on prior uptake, they are not subject to what is denoted here as the competition effect, such that if more $CO_2$ has previously been taken up by one sink compartment there is inevitably less available to be taken up by the other. In both studies significant anticorrelation was found

between the transfer coefficients across the sets of participating models and that, as a consequence, diminished inter-model variance in the time-dependent extent of drawdown of excess atmospheric $CO_2$ relative to that which would be expected in the absence of anticorrelation in the transfer coefficients.

As noted above the transfer coefficients $k_O$ and $k_T$, being based on the net transfer rates, and the extents of transfer, being the integrals of the transfer rates, are net quantities, the difference between fluxes in the opposing directions. (As $k_A$ is not a transfer

coefficient *per se*, but represents the total net flux, from the atmosphere to the WO and the TB, again normalized to excess atmospheric $CO_2$, fluctuations in $k_A$ simply represent the sum of fluctuations in $k_O$ and $k_T$, which must be considered more fundamental than those in $k_A$.) In both studies examined here fluctuations in $k_T$ greatly exceeded those in $k_O$ (and thus dominated fluctuation in $k_A$). It would seem almost certain that the large interannual fluctuations in $k_T$ in several of the models arise from externally imposed influences in processes controlling this net uptake such as temperature, cloudiness, and

precipitation, which seemingly are built into the models, or into the parent climate models, to represent variability in these processes and to examine the consequences of this variability, as opposed, to processes that would actually occur in a given hypothetical future year subsequent to the pulse emission or cessation of emissions. From an observational perspective, some sense of the temporal variability of the global rate of net drawdown of atmospheric $CO_2$ is gained from the historical time series



of fractional annual removal rate of excess $CO_2$, corresponding to $k_A$, which has ranged, over the time period 1959-2012 from as
great as 4.3% $yr^{-1}$ to as low as 0.8% $yr^{-1}$, but never negative (Raupach et al., 2014). It would thus seem that some of the models may be overestimating the effects of climate variables on the rate of drawdown of excess $CO_2$ in future climate, perhaps substantially so. In any event, given the large intra-model and inter-model fluctuations in $k_T$, it came as rather a surprise in this analysis to be able to demonstrate and quantify consistently negative inter-model anticorrelation between $k_O$ and $k_T$ across the models that participated in each of the studies.

The finding of inter-model anticorrelation between $k_O$ and $k_T$ led to determination of effect of this anticorrelation on enhancement of anticorrelation in the time-dependent extent of transfer of excess $CO_2$ into the two receiving compartments. Here the term "enhancement" is introduced to account for anticorrelation greater than that which would inevitably arise from the competition effect. A method was introduced to estimate the competition effect by randomizing the time-dependent pairs of net transfer coefficients. From comparison of the results from the model intercomparisons with those for the randomized ($k_O$, $k_T$)
pairs it was shown by the several measures of anticorrelation examined here that in the each of the two studies the anticorrelation between the amount of excess atmospheric taken up by the WO and the TB, well exceeds that due to the competition effect, leading to substantial underestimation of the inter-model diversity that had previously been inferred from these studies.

Inter-model diversity of a quantity of interest may confidently be taken as a measure of the minimum uncertainty that would attach to knowledge of that quantity, at least as represented by the set of models examined. Not uncommonly, however, inter-
model diversity is *equated* with uncertainty, e.g., in the present context, by Joos et al., (2013) and by the IPCC Fifth Assessment Report (Myhre et al., 2013). The finding here of substantial underestimation of inter-model diversity in the time-dependent extent of drawdown of excess atmospheric $CO_2$ implies a greater uncertainty in key measures of drawdown than has previously been recognized and ascribed to this quantity.

From a cumulative forcing perspective, the pertinent quantity is not the atmospheric IRF, but its integral over the time horizon of
interest. This integrated IRF, shown for the pulse emission study of J13 in their Figure 1*b* is equal to 52.4 years for the J13 multimodel mean, with standard deviation 5.5 years (10%). The difference between the violet and blue regions of **Fig. 1*c*** is the reduction in inter-model diversity (expressed as ±2 s.d.) due to anticorrelation of the transfer coefficients across the models. From integration of the upper and lower bounds of the blue shaded region in **Fig. 1*c***, which give the ±2-s.d. range of the spread in IRF that would result only from the competition effect, that is in the absence of anticorrelation of the transfer coefficients
across the models, the standard deviation of the integrated IRF increases to 8.1 years (15%), an increase of about 50%. This increase in inter-model diversity would attach also to the absolute global warming potential (AGWP) of $CO_2$, defined as the cumulative atmospheric IRF of $CO_2$ times its forcing efficiency, a quantity widely used in assessments, such as the IPCC Fifth Assessment Report (Myhre et al., 2013, p. 712), of the cumulative effects of prospective changes in emissions of $CO_2$.

The AGWP of $CO_2$ is also widely used to compare the integrated radiative forcings of non-$CO_2$ greenhouse gases (ncghgs) to
that of $CO_2$ via the global warming potential, GWP, the ratio of AGWP of ncghgs to that of $CO_2$. In the IPCC Fifth Assessment Report Myhre et al. (2013) state that the GWP has become the "default metric" for transferring emissions of different gases to a common scale, $CO_2$-equivalent emissions, observing also that the GWP for a time horizon of 100 years has been adopted as a

metric to implement a multi-gas approach to achieve specific targets. The increase in inter-model spread of the cumulative atmospheric IRF resulting from the present analysis attaches also to the GWPs and to equivalent emissions of ncghgs, affecting

the confidence that can be placed in strategies to achieve specific targets and the confidence that can be placed in replacing emissions of one ghg with another in achieving target ghg emissions. For all these reasons it seems essential that the minimum uncertainty in rate of drawdown of excess $CO_2$, as represented by inter-model diversity, be accurately known and stated. If the inter-model diversity in integrated IRF obtained here is an accurate characterization of the uncertainty in that quantity, then the uncertainties in the AGWP of $CO_2$ and in the GWPs of ncghgs would increase by like amount.

A question that naturally arises is the reason for the anticorrelation in drawdown rates across the sets of models participating in the two studies. A similar situation of anticorrelation across nominally independent models was found to exist between modeled climate sensitivity, forcing, and temperature change over the industrial era, as reported in a suite of climate model studies. This situation was initially identified (Schwartz et al., 2007) from the inter-model diversity in temperature change in climate model studies being much less than the uncertainty in forcing. Shortly afterwards Kiehl (2007) showed that this was due to inverse

correlation across the set of models between forcing and climate sensitivity. Reflecting on such anticorrelation across nominally independent models, Sherwood et al. (2020) underscore the need to identify model co-dependencies that might have the effect of diminishing uncertainty in estimates of climate sensitivity. They observe that "Modelers and process experts are aware of the historical climate record. GCM aerosol forcings might have been selected in order to match the observed warming rate over the twentieth century (e.g., Kiehl, 2007)." Sherwood et al. go onto suggest that "otherwise plausible models or feedbacks might have

been discarded because of perceived conflict with this warming rate, or aversion to a model's climate sensitivity being outside an accepted range."

It would seem that considerations such as these might apply similarly to the anticorrelation in the modeled rate of drawdown of excess $CO_2$ subsequent to pulse emission (J13) or following abrupt cessation of anthropogenic emissions (M20). Although for such $CO_2$ drawdown there is no observable quantity analogous to temperature increase over the industrial era that might directly

constrain the parameter pairs, such that if one parameter is known the other is constrained, it might nonetheless be possible that there is a perceived rate of net drawdown of excess atmospheric $CO_2$, based on historical anthropogenic emissions and observed fraction of these emissions that remain in the atmosphere (atmospheric fraction) that has served as an unrecognized constraint on parameter pairs and may have caused investigators similarly to discard otherwise plausible rates of uptake into either the ocean or the TB to obtain an atmospheric fraction as a function of time subsequent to pulse emission or abrupt cessation of emissions

that is within some range of expectation. As well, modelers may have been influenced by drawdown rates reported in prior model intercomparison studies such as Archer et al. (2009).

The present analysis has demonstrated and quantified anticorrelation across the models that participated in each of the two intercomparison studies between the transfer coefficients characterizing the net rate of transport of excess atmospheric $CO_2$ into the two receiving compartments, the WO and the TB, subsequent to the pulse emission (J13) or to cessation of anthropogenic

emissions (M20). This study has also, for each of the two studies examined, quantified the enhancement in anticorrelation in the extent of uptake in the two receiving compartments beyond that which would be expected inevitably from the competition effect alone and the resultant decrease in inter-model diversity. This study has also called attention to large intra-model temporal variability, specifically fairly high frequency changes of sign of the net transfer coefficient from the atmosphere to the TB in



some models, and to large inter-model differences in transfer coefficients, both of which tend to become rather muted when
looking at the extents of transfer, which are the integrals of the transfer rates. At the end of the day, however, this study has done
little to identify the underlying causes of these anticorrelations. Identifying these causes would, it would seem, require a much
more thorough examination of the representations of the processes governing net uptake of excess atmospheric $CO_2$ into the two
receiving compartments than can be achieved simply by intercomparisons such as the two studies examined here. Nonetheless it
would seem that if progress is to be made in representing these processes and in enhancing confidence in these representations,
such thorough inter-model comparison at the process level is required.

For the present, at minimum, it would seem incumbent upon the communities using the results of these intercomparison studies
for assessing the consequences of prospective changes in emissions or for comparing integrated forcings of $CO_2$ and other
greenhouse gases to take cognizance of the implications of anticorrelations across the models participating in the two
intercomparison studies examined here and resultant artificial decrease in inter-model diversity in these two studies.

**Competing Interests**

The author declares no competing interest.

**Acknowledgments**

I thank Andrew MacDougall for discussion and for providing the M20 data. I thank L. Kleinman and E. Lewis and one
anonymous reviewer for valuable suggestions. Much of this work was conducted while the author was at Brookhaven National
Laboratory, with support in part from US Department of Energy (Contract No. DE-SC0012704); views expressed here are those of
the author and do not necessarily reflect the views of BNL or DOE. The author declares no competing interest. This article and
corresponding preprints are distributed under the Creative Commons Attribution 4.0 License.

**Data**

Data from the ZECMIP project (MacDougall et al., 2020) are available from http://terra.seos.uvic.ca/ZEC/ . Data from the Joos
et. al. (2013) intercomparison were obtained by digitization of the figures in the published paper. Data displayed in the several
figures are presented in SI.

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
