# Peer review of "Anticorrelation of Net Uptake of Atmospheric CO2 by the World Ocean and Terrestrial Biosphere in Current Carbon Cycle Models"

_EGUsphere, 2024_

## Author Response (AR1)

Response to Associate editor decision Re: **Manuscript egusphere-2024-748**

Anticorrelation of Net Uptake of Atmospheric CO2 by the World Ocean and Terrestrial Biosphere in Current Carbon Cycle Models

by Stephen E. Schwartz

07 Sep 2024

Associate editor decision: Publish subject to minor revisions (review by editor)
by Paul Stoy

Public justification (visible to the public if the article is accepted and published):

Both Referees indicated that the manuscript is publishable subject to minor revisions and I am inclined to agree. Please upload a revised version of the manuscript that includes the improvements noted in the response letter and I would be happy to recommend it be published in Biogeosciences.

Sincerely,

Paul C. Stoy

Author Response: I am pleased at the outcome of the Review. Please see Responses below. I thank the Editor for his continued attention to this manuscript.

This document details changes to manuscript. Attached are Revised manuscript, Track Changes, and a single document comprising Abstract, 500-word summary, and Graphical Abstract.

**Summary of modifications to manuscript egusphere-2024-748**

Modifications to manuscript are as presented in responses to reviewers, below. The Track Changes looks much worse than is actually the case mainly because of mods to conform to Copernicus style. As I expect the Editor recognizes, green in the track changes merely denotes material that is moved from one place to another.

There is substantive revision in response to comment of Reviewer 1 commencing at line 490 of the Revision, beginning "A *sine qua non* . . . " I explicitly acknowledge MacDougall in his capacity as reviewer for stimulating my thoughts along these lines.

Some text has been rearranged. Figures have been replaced to conform to Copernicus style "(a)" etc, Numerous changes are made to conform to Copernicus style. References have been revised to conform to Copernicus style.

I have provided a graphical abstract, which is extracted from Fig. 1 of the manuscript.

- - -

**Response (2024-0827) to** RC1: 'Comment on egusphere-2024-748', Andrew MacDougall, 30 May 2024

I appreciate the Reviewer's attention to the manuscript and generally positive and helpful comments. I am especially pleased that the Reviewer finds the analysis convincing and the paper well written.

**Review**: Anticorrelation of Net Uptake of Atmospheric CO2 by the World Ocean and Terrestrial Biosphere in Current Carbon Cycle Models

*Overall Evaluation:*

The manuscript used results from two prominent model intercomparison projects of Earth system models to show that uptake of carbon by the terrestrial biosphere and ocean are anti-correlated, beyond the degree one would expect from conservation of mass. The analysis is convincing and the the paper is well written. I believe that the paper can be published pending minor revisions.

*General Comments:*

(1) It would be helpful in the Introduction to briefly describe the principle underlying processes that drive ocean (air-sea gas exchange) and land (photosynthesis and respiration) carbon uptake .

A short para is added, line 34-42,  describing the underlying processes. I thank the reviewer for the suggestion

*Specific Comments:*

Line 82: Please re-write sentence for clarity.

The sentence is divided into two. (line 87 in the revision)

Line 87: J13 has also been used to tune the FAIR climate emulator (e.g. Smith et al. 2018), a very commonly used climate emulator.

I thank the Reviewer for calling attention to this. A sentence to this effect has been added. Line 92

Figure 1 (and other in the same format): The extra lines on panel d and e are visually distracting. Though they do add scientific value, the lines obscure the main point of these panels.

The reviewer notes that the curves (thin curves, green, cyan, brown) that show indivdual instances add scientific value, but he finds them distracting. I would prefer to leave them in the figure as they give a sense of the extent of fluctuations of individual pairings of sets of ($k_O$, $k_T$) pairs. I think the reader will figure it out.

Line 231 to 232: Did you intend to refer to Figure 3 here?

Yes. Thank you. Corrected.

Line 269: Note quite correct. Simulations stopped when 1000PgC of cumulative emissions was diagnosed. By coincidence this is almost the same point in time that most models reach 2X CO2.

Corrected

Figure 4: Metadata for this figure is visible.

Deleted

Line 429: Fire is also a major contributing factor to inter-annual variability in the land carbon uptake (e.g. Di Virgilio et al. 2019).

Added (but without citation)

Line 454 to 475: Far too many abbreviations in these two paragraphs. I recommend writing all of them out except 'GWP', which is a common abbreviation.

I have looked over these two paras. The first para (beginning line 545 in the original, 471 in the revision) introduces (and defines) only the abbreviation AGWP, in addition to well known abbreviations IRF and IPCC.

The second para (beginning line 464 in the original, 481 in the revision) introduces (and defines) only ncghg and introduces ghg, in addition to the AGWP, and IPCC previously introduced and GWP, which it specifically defines.

On re-reading these paras, I feel that they do not overly tax the reader and suggest leaving as is.

Line 486 to 496: I suspect tuning models to match the Keeling curves may be the source of the anti-correlation. Virtually all ESM try to replicate the curve as a test of their carbon-cycles.

This comment caused me to think more about the pertinent paragraph. In the revision I avoid the use of the word "tuning" and the phrase "Keeling curve" suggested by the Reviewer. But I note explicitly that in historical runs satisfying the observational requirement of growth in atmospheric CO2 being equal to the difference between emissions and the sum of uptake into the ocean and the TB sets up a potential for anticorrelation in models. I thank the Reviewer for the comment.

I also tightened that paragraph and re-arranged and removed some materials in the section.

*References:*

Smith CJ, Forster PM, Allen M, Leach N, Millar RJ, Passerello GA, Regayre LA. FAIR v1. 3: a simple emissions-based impulse response and carbon cycle model. Geoscientific Model Development. 2018 Jun 18;11(6):2273-97.

Giovanni Di Virgilio, Jason P Evans, Stephanie AP Blake, Matthew Armstrong, Andrew J Dowdy,Jason Sharples, and Rick McRae. Climate change increases the potential for extreme wildfires. Geophysical Research Letters, 46(14):8517–8526, 2019.

- - -

**Response (2024-0827) to RC2 Anonymous Referee #2, 22 Aug 2024**

Per se interesting study. Not so easy to read. I have focussed on the part presenting the first model inter comparison study only. The results seem similar though.

Main point of manuscript that there is a trade-off in carbon cycle / earth system model simulations (results from two model inter comparison experiments are being analysed) between uptake of the anthropogenic perturbation of atmospheric carbon by oceans versus land pools is convincing  (primarily figures 1 a-c and figure 2a).

I appreciate the Reviewer's attention to the manuscript. I am especially pleased that the Reviewer finds the argument convincing.

Author suggests this is due to

'… compensating treatments in the several models of the processes that govern uptake of excess $CO_2$' (excess $CO_2$ defined as 'atmospheric $CO_2$ above pre-industrial amount') and not to 'competition' between the two sinks defined as

 '*competition between the two sinks for the excess atmospheric $CO_2$ remaining from the pulse at a given time subsequent to the pulse; if, at a given time, more of this excess $CO_2$ has previously been drawn down into one compartment, there would be less $CO_2$ available to be drawn down into other compartment*'

To disentangle this question the author states / uses

'*As these net transfer coefficients are not dependent on prior uptake, they are not subject to what is denoted here as the competition effect, such that if more $CO_2$ has previously been taken up by one sink compartment there is inevitably less available to be taken up by the other*'

In the above text and quotations from the manuscript the reviewer accurately summarizes the line of reasoning.

I am not sure / convinced this statement is correct (e.g. Revelle effect, land vegetation carrying capacity, land vegetation $CO_2$ fertilization). While I may well be missing something (if so apologies) I do think the author needs to explain convincingly this claim in a revised version.

The instantaneous (temporally local) net uptake coefficients (net uptake per excess atmospheric $CO_2$, above preindustrial, depends only on instantaneous rates of uptake into the two compartments. The several processes noted by the Referee affect uptake by the respective compartments but not the "other" compartment. The Revelle effect affects the rate of uptake into

the ocean but not the rate of uptake into the TB; the land carrying capacity and fertilization affect the rate of uptake into the TB but not the rate of uptake into the ocean. There is no "cross talk" whereby Revelle effect might affect the rate of uptake into the TB, or where carrying capacity or fertilization could affect the rate of uptake into the ocean.  These processes would be expected to have affected the amount of excess atmospheric CO2 present at a given time, resulting in what I have denoted the compensation effect leading to autocorrelation in the amount of CO2 taken up in the two compartments (and indeed this compensation effect is quantified in the manuscript). However examination of the temporally local rates upon temporally local amount of excess CO2 obviates this concern.

Author states *'compensating treatments' … more concerning ….*

While the detected anti-correlation may be of some concern (some implicit constraint on the sum of ocean and land uptake) in my view (looking e.g. at Figure 1) at the 2013 stage off earth system modelling (and probably still in 2024)  a much larger concern is  that the spread across models of the predicted land carbon uptake of 'excess atmospheric carbon' is very large - ranging from nearly zero to very susbtantial uptake. Essentially we are not really in a position to predict how much / whether land carbon pools will have increased or decreased by the year 2100. Thus probably at the current stage of knowledge the issue highlighted by the author of the study is probably a second order problem only.

The reviewer hits the nail on the head. Looking at Figure 1c (for the Joos intercomparison), the spread in atmospheric IRF at 100 years is about a factor of 2, 0.3 to 0.6 or maybe a factor of about 1.5 about the average. But looking at Figure 1b for the terrestrial IRF, ignoring the one outlier, the range is a factor 4, 0.1 to 0.4 or a factor of 2 about the average. Part of the motivation for the present manuscript is to call attention to this and to try to identify the reason(s) for this. A major contributor is the anticorrelation under examination here.

There are of course other intrinsic reasons for the spread in IRFs of the Ocean and TB uptake that are rooted in the treatments of the uptake processes in the several models. The point of an intercomparison is to accurately assess the spread, not necessarily to identify the reasons for the spread, but importantly to give some measure of the confidence that can be placed in model results based on spread. This study shows that the spread in the atmospheric IRF, the key quantity that is a measure of that spread, is artificially reduced because of the anticorrelation identified and quantified here.

**Minor comments**

*Figure 1*

1. sometimes the same symbol is used for two different models (e.g. red dots, blue diamonds). Please use different symbols for different models- such that anti-correlation can be spotted / checked visually. Where are the black triangles in figure a ?

I struggled with the line and symbol codes. In the analysis of the J13 results I used the same codes as in the original paper, which used these codes to distinguish model type (comprehensive Earth system models, Earth system models of intermediate complexity, and box models). To

distinguish the symbols it seems that I made the symbols too large so that they obscure the underlying symbols. I have reduced the symbol size in Figures 1 and 3 to help better identify the models across the panels. I don't think any approach is "perfect". I do use vector graphics so that the interested reader can zoom in on the screen.

The black triangles in figure 1a are partly hidden by the green circles and the red diamonds; in the revision they are more discernable toward the left and the right.

2. please describe in more detail how figures d and e have been calculated

The quantities shown in Figure 1d are the slopes of the regression fits of $IRF_O$ vs $IRF_T$ as shown in a single example in Figure 2a, all as a function of time $t$, as stated at lines 100-101. Similarly Figure 3d for k.

3a. is the regression made on ensemble mean ?

No. The regression is made in the usual way (see citation to Press et al., 1997), minimizing the sum of squares of the departure of the y distances from the regression line, where the sum is taken over the several models.

3b. is the regression repeated on IRF(1},..., IRF(n)} for n=1, 2, .. 99 ?

The quantity plotted in Fig. 1d is {$dIRF_O(n)/dIRF_T(n)$} for n (year since pulse) =1, 2, .. 99, where the notation $dIRF_O(n)/dIRF_T(n)$ denotes the slope of the regression line obtained by least squares fit of the points $IRF_O(i)$ vs $IRF_T(i)$ for all models $i$ = 1...14. The text refers frequently to "regression slope"; the axis label e.g. Figure 1d, $dIRF_O(n)/dIRF_T(n)$ denotes that slope.

4. why notation dIRF  - i.e. why 'd' ? is it not just the the regression slope (i.e. not small increments)

Yes, it is slope, as in the slope of a line, dy/dx. $dIRF_O/ dIRF_T$ is a very different quantity from the quantity without the d,  $IRF_O/ IRF_T$.

5. please give more detail.

The procedure is presented in detail at lines 138-145 for the IRFs and 156-180 for the net transfer coefficients. The procedure for the randomized pairs is presented in detail at lines 199-208 and 224-258.

6. it would be helpful to see all correlation time-series [i.e. for each model in the model suite]

If I correctly understand the referee's query, I would respond that a single model is not correlated. What is shown is time series of the correlation slope (for the set of models) e.g. fig 1d and correlation coefficient (for the set of models) r, e.g., fig 1.

7. Good to correct a few typos (e.g. 'that that..')

Good catch. I tried to catch them all and had others read the manuscript for that purpose as well as substance; inevitably some still typos seem to have persisted. I have looked the manuscript over again

---

## Author Response (AR2)

12 Sep 2024

Minor change. I added identifying arrows in the graphical abstract.

Stephen E Schwartz

As approved:

[Figure]

As revised: